# An In Situ Incorporation of Acrylic Acid and ZnO Nanoparticles into Polyamide Thin Film Composite Membranes for Their Effect on Membrane pH Responsive Behavior

**DOI:** 10.3390/membranes11120910

**Published:** 2021-11-23

**Authors:** Kgolofelo I. Malatjie, Bhekani S. Mbuli, Richard M. Moutloali, Catherine J. Ngila

**Affiliations:** 1Department of Chemical Sciences, University of Johannesburg, Johannesburg 2028, South Africa; malatjiekgolofeloinna@gmail.com (K.I.M.); moutlrm@unisa.ac.za (R.M.M.); jcngila2015@gmail.com (C.J.N.); 2Department of Science and Innovation/Mintek Nanotechnology Innovation Centre-Water Research Node, University of Johannesburg, Doornfontein, Johannesburg 2028, South Africa; 3Institute for Nanotechnology and Water Sustainability, College of Science, Engineering and Technology, University of South Africa, Johannesburg 1710, South Africa

**Keywords:** polyamide, interfacial polymerization, acrylic acid, zinc oxide, thin film

## Abstract

This paper focuses on an in situ interfacial polymerization modification of polyamide thin film composite membranes with acrylic acid (AA) and zinc oxide (ZnO) nanoparticles. Consequent to this modification, the modified polyamide thin film composite (PA–TFC) membranes exhibited enhanced water permeability and Pb (II) heavy metal rejection. For example, the 0.50:1.50% ZnO/AA modified membranes showed water permeability of 29.85 ± 0.06 L·m^−2^·h^−1^·kPa^−1^ (pH 3), 4.16 ± 0.39 L·m^−2^·h^−1^·kPa^−1^ (pH 7), and 2.80 ± 0.21 L·m^−2^·h^−1^·kPa^−1^ ^1^ (pH 11). This demonstrated enhanced pH responsive properties, and improved water permeability properties against unmodified membranes (2.29 ± 0.59 L·m^−2^·h^−1^·kPa^−1^, 1.79 ± 0.27 L·m^−2^·h^−1^·kPa^−1^, and 0.90 ± 0.21 L·m^−2^·h^−1^·kPa^−1^, respectively). Furthermore, the rejection of Pb (II) ions by the modified PA–TFC membranes was found to be 16.11 ± 0.12% (pH 3), 30.58 ± 0.33% (pH 7), and 96.67 ± 0.09% (pH 11). Additionally, the membranes modified with AA and ZnO/AA demonstrated a significant pH responsiveness compared to membranes modified with only ZnO nanoparticles and unmodified membranes. As such, this demonstrated the swelling behavior due to the inherent “gate effect” of the modified membranes. This was illustrated by the rejection and water permeation behavior, hydrophilic properties, and ion exchange capacity of the modified membranes. The pH responsiveness for the modified membranes was due to the –COOH and –OH functional groups introduced by the AA hydrogel and ZnO nanoparticles.

## 1. Introduction

Polyamide thin film composite (PA–TFC) membranes have been reported to possess a high capacity for the removal of heavy metals, dissolved salts, and organic solutes from wastewater [1,2,3]. Notably, heavy metal rejection, water permeability, and fouling resistance are controlled by the PA ultra-thin layer of these TFC membranes [4,5]. However, TFC membranes are challenged by their low water permeability. This is because of the highly crosslinked structure of the PA active thin film layer, and consequent increased fouling [6]. Therefore, to offset these challenges, extensive efforts are being made to modify the active polyamide thin film of the TFC membranes. Various researchers are modifying the polyamide thin film with suitable additives, such as nanoparticles, to improve the membranes’ hydrophilicity. Such newly incorporated properties can inherently translate into enhanced water permeability and fouling resistance of the modified membranes [7].

Nanoparticles, such as ZrO_2_, Al_2_O_2_, SiO_2_, Fe_3_O_4_, ZnO, and TiO_2_, have been reported to impart hydrophilicity onto modified PA–TFC membrane surfaces [8,9,10,11]. As such, their incorporation into PA–TFC membranes improves the water permeability and fouling resistance of PA–TFC membrane surfaces. Consequently, inorganic nanoparticles have found widespread application in the modification of PA–TFC membranes suitable for water treatment [12,13]. Furthermore, these inorganic nanoparticles have been extensively applied as nano-adsorbents in order to remove heavy metal ions from wastewater. This is because of their high surface area, high activity, and enhanced structural stability [14,15,16,17,18,19,20]. For instance, Al-Hobaib et al. reported that ZnO nanoparticles enhanced hydrophilic properties of reverse osmosis (RO) PA–TFC membranes. As such, the water permeate flux of the modified membranes was enhanced, especially at a lower concentration of 0.5 wt.% ZnO nanoparticles [21].

In particular, Ag-ZnO/PA–TFC membranes have demonstrated higher flux recovery and fouling resistance against 2-chlorophenol (2-CP) and 2,4-dichlorophenol (2,4-DCP) pollutants when compared with unmodified PA–TFC membranes. These enhanced properties were due to the improved hydrophilic properties that inherently improved water permeation, rejection, and fouling resistance properties of the modified PA–TFC membranes [22,23,24]. In addition, Namvar-Mahboub and Pakizeh demonstrated that nanofiltration (NF) TFC membranes supported on polyether amide (PEI/amino-functionalized) silica nanocomposites improved modified membranes’ water permeate flux and their oil rejection capabilities [23]. Furthermore, the ZnO nanoparticles demonstrated capabilities towards the removal of heavy metals through an adsorption mechanism in water treatment [24].

Acrylic acid (AA) hydrogels, on the other hand, have been shown to introduce pH responsive properties and enhance hydrophilicity of modified PA–TFC membranes [25,26,27,28,29]. Therefore, such properties on modified membranes can inherently reduce fouling propensity as a result of the external stimuli responsive properties [25]. External stimuli responsive hydrogels, also known as “smart” polymers, can respond to externally applied stimuli such as temperature, light, pH, and pressure [26,27,28,29]. Therefore, incorporating these “smart” polymers into PA–TFC membranes can significantly influence membrane properties, such as water permeability, solute rejection, and fouling resistance.

As such, acrylic acid hydrogels have demonstrated pH responsive properties, shown by swelling and shrinking behavior under varied pH conditions [30]. This means that AAs are pH sensitive hydrogels that can shrink at lower and swell at higher pH conditions. Such properties can translate into a “gate effect” within a membrane matrix. The “gate-effect” properties translated into membranes can effectively lead to a higher passage of water molecules and dissolved solutes under acidic conditions and restricted passage under basic conditions [31,32,33,34]. This is attributable to the ionizable –COOH functional groups of the hydrogels. Thus, in this research work, the Pb^2+^ pollutant was utilized as a model contaminant to test the strength of the pH responsive “gate effect” of the modified membranes. The aim was not only to remove Pb^2+^ through size exclusion, but to clearly understand the adsorption–desorption mechanism as a result of the pH responsiveness of the modified membranes. 

Notably, the switchable behavior of the hydrogel can render the modified membrane “self-cleaning” because of the adsorption and desorption properties which result of simply switching the pH conditions of feed water solution [35]. For example, the NIPAmA copolymers and AA hydrogels have demonstrated pH responsiveness in modified PA–TFC membranes [33]. These modifications have resulted in enhanced water permeability, solute rejection, and fouling resistance of the pH-responsive PA–TFC membranes [33,34]. However, salt rejection capabilities of these modified membranes were reduced due to fewer negative charges available on the membrane surfaces under neutral and basic conditions [36]. 

Notably, most researchers usually graft the acrylic acid onto commercial thin films as brushes on commercial PA–TFC membranes. For example, pH-responsive poly(acrylic acid) (PAA) brushes were grafted onto the surface of a commercial TFC-PA membrane using surface-initiated atom transfer radical polymerization. The NIPAmA copolymers and AA hydrogels were grafted on commercial PA–TFC membranes. However, the approach of this research was to execute an in situ incorporation of the AA hydrogels during the interfacial polymerization process. Such an approach was envisaged to stabilize the AA and ZnO nanoparticles into the polyamide thin film matrix. Furthermore, the grafted PA–TFC membranes were often observed to exhibit decreased water permeability because of the extra layer on the polyamide thin film. Therefore, in this work, an in situ interfacial polymerization was pursued. 

Herein, a detailed report of an in situ interfacial modification of PA–TFC membranes with AA hydrogels coupled with ZnO nanoparticles is presented. This approach to the modification of the TFC layer is novel and is, to the best of our knowledge, reported here for the first time. The novelty consists in the in situ incorporation of the AA hydrogels coupled with ZnO nanoparticles to introduce pH responsiveness into the PA–TFC membranes. In particular, a detailed pH response and its effect on membrane swelling, water contact angle, gating effect (i.e., Pb (II) rejection and pure water flux), and ion exchange are reported. A comparative study of PA–TFC membranes modified with different concentrations of ZnO nanoparticles coupled with AA hydrogels are reported to demonstrate the improved pH responsive behavior.

## 2. Materials and Methods Experimental

### 2.1. Materials

Acrylic acid (AA, 99.0%), sodium dodecyl sulphate (SDS, 98.5%), hexane (98.0%), 1,3,5-benzenetricarbonyl trichloride (trimesoyl chloride, TMC) (98.0%), m-phenylenediamine (MPDA, 98.0%), hexane (98.0%), zinc oxide (ZnO) nanoparticles (<50 nm particle size (BET), >97%), and ammonium chloride (99.5%) were purchased from Sigma-Aldrich (Darmstadt, Germany) as analytical grade reagents. These reagents were all utilized without any further purification. Ultrafiltration (UF) polyethersulfone (PES) (PES-Radel 300, 5.00 kDa) membranes were purchased from Microdyn Nadir (Johannesburg, South Africa).

### 2.2. Membrane Preparation

The polyamide thin film composite (PA–TFC) membranes were synthesized through a modified interfacial polymerization process, as previously reported by Xie et al. [37] and shown in the schematic diagram in Figure 1. Typically, UF-PES support membranes were pre-treated through soaking in sodium dodecyl sulphate (0.50% *m/v*) overnight. After which, the membranes were washed with deionized water for 1 h and allowed to dry under a fume hood for 2 h. The membranes were then immobilized onto a glass plate using double-sided tape together with paper tape. A series of percentage mass per volume (% *m/v*) concentration aqueous solutions were prepared as shown in Table 1. The newly prepared aqueous solutions were then decanted onto the surface of the different immobilized UF-PES membranes. 

The membranes were allowed to soak for 30.00 min, after which, excess aqueous solution was removed from the UF membrane using an air knife. After removing the excess aqueous solution, trimesoyl chloride (TMC, 0.40% *m/v*) in hexane (100 mL) was poured onto the surface of the membranes and left to stand for 60 s. Afterwards, the membranes were cured in an oven for 15 min at 60 °C in order to allow for further crosslinking of the polyamide thin film. After curing, the membranes were washed with deionized water to remove unreacted monomers. Finally, the newly synthesized membranes were stored in a refrigerator until they were used. The incorporation of zinc oxide (ZnO) nanoparticles and acrylic acid (AA) onto the polyamide thin film composite (PA–TFC) membranes is shown in Figure 1. 

### 2.3. Characterization of Membranes

#### 2.3.1. Contact Angle Analysis

The hydrophilicity of the membranes was measured and collected using a Dataphysics Contact Angle instrument SCA 20(Dataphysics Instruments GmbH, Filderstadt, Germany). A sessile drop of water (0.50 μL), at the rate of 2.00 μL/s, was used when performing the contact angle measurements for all at three different pH levels (3, 7, and 11). For each membrane, six measurements at different spots of the membranes were recorded and averaged to obtain average contact angles.

#### 2.3.2. Attenuated Total Reflectance-Fourier Transform Infra-Red Spectroscopic Analysis

Attenuated Total Reflectance-Fourier Transform Infra-Red (ATR-FTIR) spectroscopy was used for functional groups analysis of all modified and unmodified membranes. The spectra were collected using an ATR-FTIR Spectrometer (Spectrum 100, Perkin Elmer, Llantrisant, UK). The characterization of the TFC membrane samples were carried out using a diamond–germanium–selenium crystal with incident angle of 45°. The spectra were averaged over 12 scans in a range from 650 cm^−1^ to 3500 cm^−1^.

#### 2.3.3. Scanning Electron Microscopic Analysis

A VEGA3 TESCAN (TESCAN, Brno, Czech Republic) scanning electron microscope (SEM) was used to determine the surface morphology of the modified and unmodified PA–TFC membranes. The membranes were mounted, and carbon coated on a carbon tape before being analyzed.

#### 2.3.4. Atomic Force Microscopic Analysis

Atomic force microscopy (AFM) analyses were carried out to quantitatively study the surface roughness of the PA–TFC membranes. This was performed using a Nanoscope 3D Multimode from Veeco Instruments Inc (Plainview, NY, USA). This instrument utilized SNL cantilevers (Veeco) with a spring constant of 0.12 Nm^−1^ through the contact mode in dry air. This was performed at appropriate magnification and accurate focusing for better viewing of the specimens, after drying the samples for 12 h in a vacuum oven.

#### 2.3.5. Thermal Gravimetric Analysis

Thermal gravimetric analysis (TGA) of the PA–TFC membranes was executed in order to investigate the thermal stability of the membrane materials. The analysis was carried out using TGA 4000 (Hitachi, Tokyo, Japan). The samples were heated under nitrogen (N_2_) gas from 25 °C up to 900 °C, at the rate of 10 °C/min.

#### 2.3.6. X-ray Diffraction Spectroscopic Analysis

The crystallinity of the PA–TFC was determined by X-ray diffraction (XRD) spectrometric analysis. The analysis was carried out using PANalytical XPERT-PRO Diffractometer (PANalytical, Malvern, UK) using CU Kα (λ = 1.54) between 2ϴ values of 2–40.

### 2.4. Membrane Performance Evaluation

#### 2.4.1. Membrane Water Permeation and Solute Rejection Study

Water permeation and solute rejection properties of the membranes were studied by using the Sterlitech HP 4750(Auburn, AL, USA) stirred cells. The PA–TFC membranes were all cut into equal sizes of 38 mm in diameter and compressed in a membrane disc into a stirred cell system in a presence of a magnetic stirrer. The stirred cell was pressurized using nitrogen gas. Water permeation studies were carried out for the unmodified and modified membranes using deionized water solution of three different pH conditions (3, 7 and 11). The basic and acidic feed water solutions were obtained by adjusting feed water solutions with NaOH (1.00 M) and H_2_SO_4_ (1.00 M), respectively. The feed water solutions of deionized water adjusted to different pH were measured at 400 kPa.

The rejection of Pb^2+^ ions was performed by preparing a stock solution of 1.00 g/L using lead (II) nitrate (Pb(NO_3_)_2_). The solutions were adjusted to three different pH conditions (3, 7, and 11). The filtrate was collected at a known volume, pressure, and time. The final concentration of lead found in the permeate water solution was determined using an Atomic Absorption Spectroscopy (AAS, Medbourne, Victoria, Australia) instrument. The Pb(NO_3_)_2_ solutions were diluted to a 1.00:100 dilution factor.

Water permeate flux (*J_w_*) was determined using Equation (1) [38]:(1)$$J_{w}=\frac{V}{tA}$$
where *V* is the volume (m^3^) of the permeate water, *t* is permeation time (s), and *A* is membrane surface area (m^2^).

The rejection percentage of solutes was calculated using Equation (2) [38]:(2)$$R=\left[1-\left(\frac{C_{p}}{C_{f}}\right)\right]\times 100\%$$
where *C_p_* and *C_f_* are the concentrations of dissolved solutes in the permeate and feed water solutions, respectively.

#### 2.4.2. Membrane Swelling Studies

The membrane swelling behavior of the PA–TFC membranes at different acrylic acid and ZnO nanoparticle concentrations was studied at three different pH conditions. Equal sized membranes (3.00 mg) were immersed in deionized water of different pH levels and stirred for 24 h. Afterwards, the membranes were gently wiped to remove excess water content and then weighed in a balance. The membranes were then put into an oven to dry at 60 °C for another 24 h and weighed again. The swelling ratios were determined using Equation (3) [39]:(3)$$\%SR=\frac{W_{sw}-W_{dr}}{W_{sw}}\times \;100\%\;$$
where *SR* represented the swelling percentage ratio, *W_sw_* was the swollen membrane mass (g), and *W_dr_* was the dry membrane mass (g).

#### 2.4.3. Ion Exchange Capacity Analysis

Ion exchange capacity (IEC) of the membranes was determined by a method previously reported by Sinha et al. [39]. Typically, the PA–TFC membranes were all cut into equal sizes and kept in a NaCl solution (50 mL, 2.00 mol/L) for 24 h with a temperature kept at 30 °C. This was performed in an orbital shaker to complete substitution of H^+^ by Na^+^ ions. The membranes were then taken out of the NaCl solution to dry them for another 24 h at 30 °C. Consequently, the dry weight of the membranes was obtained. A volume of 25.00 mL from each solution was then titrated with 0.01 M of NaOH solution using phenolphthalein as an indicator. The IEC of the membranes was then calculated using Equation (4) [39]:(4)$$IEC=\frac{0.01\times 1000\times V_{NaOH}}{W_{d}}\;$$
where *V_NaOH_* was the volume of NaOH solution (L) used for titration and *W_d_* was the mass of the dry membrane.

## 3. Results and Discussion

### 3.1. Attenuated Total Reflectance-Infra-Red Spectroscopic Analysis

Figure 2 showed the ATR-FTIR spectra of the UF-PES support membrane, unmodified PA–TFC membrane, and membranes modified with AA and ZnO/AA. For the membranes modified with AA, two concentrations of 1.00% and 1.50% AA were chosen as constant concentrations. This means that at 1% *m/v* AA or 1.5% *m/v* AA concentration, the concentration of ZnO nanoparticles were gradually increased to study the impact of the ZnO nanoparticles. As such, for these AA concentrations, increasing concentrations of the ZnO nanoparticles were introduced and studied to investigate the effect of the increasing nanoparticle concentration, as shown in Table 1. These concentrations of AA were determined after a series of optimizations of AA concentrations on the membranes using rejection capacity and water permeation as parameters for determining the appropriate loading concentrations. 

Based on Figure 2, the absorbance peaks at 1485 cm^−1^ were associated with the aromatic rings of the UF-PES membrane support. The absorbance peak at 1141 cm^−1^ was assigned to the aryl-O-aryl (–C-O-C) stretch of the UF-PES membrane support (Figure 2b,d). The broad peaks ranging between 1359 cm^−1^–1289 cm^−1^ and 1113 cm^−1^–1029 cm^−1^ represented the symmetric and asymmetric stretching of the sulfone (–SO_2_) groups, respectively [40,41]. The absorbance peak observed at 1611 cm^−1^ was assigned to the –C=O carbonyl functional group indicative of the amide I (–N-H) group [5,42]. This peak appeared at the regions ranging between 3367–1415 cm^−1^ [43] (Figure 2a,c).

The absorbance peak observed at 1201 cm^−1^ represented the –C-N (amide III) stretching of the –CO-NH primary amide (Figure 2b,d). The presence of these amide groups both indicate the presence of a polyamide layer on the TFC membrane. The absorbance peaks found at 1723 cm^−1^ were assigned to the –COOH functional groups that were indicative of the unreacted acyl chloride functional groups for the unmodified membranes and the presence of –COOH functional groups from acrylic acid for the modified membranes. The absorbance peaks at 1307 cm^−1^ were obtained because of the presence of the –C-O functional groups that validated the presence of –COOH, –OH, and –C=O functional groups. The –OH functional group peak was also observed as a broad peak around 3402 cm^−1^ of the spectra. This functional group also overlapped with the –N-H functional groups from the MPDA monomer. The –C-H functional group of the aromatic rings can also be observed around the same region as that of the –OH functional broad peaks [44]. Small absorption peaks were also found at 2918 cm^−1^, representing the –C-H alkyl functional groups of the AA polymer [43]. Even though there were minimal contrasting differences between the ATR-FTIR spectrum of the PA–TFC membranes modified with 1.00% and 1.50% AA and their corresponding ZnO/AA additives, there were slight shifts in the absorbance peaks even though these shifts were minor. These slight band shifts were indicative of the interaction between the ZnO nanoparticles and the acrylic acid with the amine groups from the MPDA monomers. Consequently, these ZnO nanoparticles and AA additives were interacting with the amide I functional groups of the modified membranes. This kind of behavior was not observed with the unmodified membranes and membranes modified with 1.00% and 1.5% AA hydrogels (Figure 2b,d). This was confirmed by the fact that the amide I functional group of the unmodified membranes and AA hydrogel-only modified membranes was weakest, showing that the incorporation of the AA and ZnO nanoparticles affected the polyamide thin film formation during the interfacial polymerization reaction.

### 3.2. Surface Morphological Study of Membranes

#### 3.2.1. High-Resolution Scanning Electron Microscopic (HR-SEM) Analysis

The HR-SEM was used to determine the surface morphology of the PA–TFC membranes, as shown in Figure 3. The top surface morphology of the modified membranes showed uniformity when the ZnO/AA nanoparticles had been incorporated at different concentrations in comparison to unmodified membranes. These modified membranes were even smoother than the membranes modified with AA only. The membranes modified with 1.00% AA, 0.50:1.00% ZnO/AA, 1.00:1.00% ZnO/AA, and 1.00:1.50% ZnO/AA showed a more even distribution of the ZnO nanoparticles on the surface than the other modified membranes (Figure 3c–e). The membranes modified with 0.50:1.50% ZnO/AA, 1.00:1.50% ZnO/AA, and 1.50:1.50% ZnO/AA, as observed with the membranes modified with 0.50:1.00% ZnO/AA, were found to display an even distribution of the nanoparticles (Figure 3c,g–i). An uneven distribution of the ZnO nanoparticles could indicate agglomeration of the ZnO nanoparticles, as was shown in Figure 3i, which could negatively affect the performance of the modified membranes.

#### 3.2.2. Atomic Force Microscopy Analysis

Atomic force microscopy was used to determine the roughness of the membrane surface and evaluate the effect of ZnO nanoparticles on the membrane surface structure (Figure 3, Figure A1 (Appendix A and Table 2)) [20,21]. It was noted that the roughness of both the membranes modified with 1.00% AA and 1.50% AA and ZnO nanoparticles were highly dependent on ZnO nanoparticle concentration. Generally, an increase in ZnO nanoparticle concentration resulted in a decrease in mean roughness of the modified membranes. For example, it was observed that the membranes modified with 0.50:1.00% ZnO/AA and 0.50:1.50% ZnO/AA had the highest mean roughness when compared to the other ZnO/AA modified membranes, as shown in Table 2. This observation confirmed the observations made from the top SEM images (Figure 3) which suggested that the rough surfaces were consistent with an even distribution of the ZnO nanoparticles. Therefore, it was inferred that an increase in ZnO nanoparticles was directly proportional to the roughness of the modified membranes.

The mean roughness (R_a_) of the unmodified membranes was found to be 183.11 nm. The membranes modified with 1.00% AA had an R_a_ of 77.63 nm and the membranes modified with 1.50% AA had an R_a_ of 50.13 nm. The latter showed a lower R_a_, thus indicating that the increment of the hydrogel concentration improved the smoothness of the modified membrane surfaces. However, the incorporation of ZnO nanoparticles to these membranes led to an increase in roughness, apart from the membranes modified with 1.00:1.00% ZnO/AA that had an R_a_ of 56.98 nm. This was also demonstrated by membranes modified with 1.00% AA, 0.50:1.00% ZnO/AA, 1.00:1.00% ZnO/AA, and 1.50:1.00% ZnO/AA, with R_a_ values of 77.63 nm, 218.80 nm, 56.98 nm, and 60.19 nm, respectively. This clearly demonstrated an increase in smoothness with an increase in ZnO nanoparticles loading. The membranes modified with 1.50% AA, 0.50:1.50% ZnO/AA, 1.00:1.50% ZnO/AA, and 1.50:1.50% ZnO/AA had an R_a_ of 50.13 nm, 208.80 nm, 146.56 nm, and 56.23 nm, respectively. This shows similar behavior as the former. Furthermore, for all the membranes with ZnO/AA, it was distinguished that the incorporation of ZnO nanoparticles increased the surface roughness of the membranes when compared to unmodified membranes and membranes modified with AA hydrogels only. Generally, the membranes modified with 1.00% AA, 0.50:1.00% ZnO/AA, 1.00:1.00% ZnO/AA, and 1.50:1.00% ZnO/AA had slightly smooth surfaces when compared to membranes modified with 1.50% AA, 0.50:1.50% ZnO/AA, 1.00:1.50% ZnO/AA, and 1.50:1.50% ZnO/AA. However, the membranes modified with 0.50:1.00% ZnO/AA and 0.50:1.50% ZnO/AA from each category gave the highest rough surfaces and the membrane modified with 0.50:1.00% ZnO/AA was found to be the roughest membrane. Some studies have shown that rough surfaces result in membrane fouling, as they contain “valleys” or “troughs” that promote foulants entrapment [45]. However, the same rough surfaces are beneficial as they can lead to high water adsorption sites, thus promoting higher water permeability in modified membranes [45].

Smooth surfaces, on the other hand, have been reported to reduce membrane fouling propensity as well as adsorption sites for water molecules [46,47]. Nonetheless, hydrophilic membrane surfaces have been shown to be less prone to membrane fouling propensities. This is a result of the reduction of binding sites for foulants entrapment on the membrane surfaces [48,49]. Therefore, in this research work, it was noted that the membranes modified with 0.50:100% ZnO/AA and 0.50:1.50% ZnO/AA possessed the highest roughness, hydrophilicity, and permeability, as shown in Figure 4 and Figure 5. The former membrane showed even higher roughness and hydrophilicity when compared to the rest of the modified membranes. The hydrophilicity of the modified membranes was influenced by the incorporation of the ZnO nanoparticles. For example, Hobaib et al. reported that an optimum concentration of ZnO nanoparticles was 0.50 wt.% when performing an in-situ interfacial polymerization modification. They demonstrated that with this amount of nanoparticle loading, water permeability and hydrophilicity were greatly improved [20]. Hobaib and the co-workers reported that an increase in ZnO concentration of more than 0.50% decreased the water permeability drastically [20]. These observations were also demonstrated with the modified membranes prepared for the present work, as shown in Figure 4 and Figure 5.

Peak densities (P_n_) were found to be directly proportional to the average roughness (R_a_), as an increase in P_n_ showed an increased in R_a_ (Table 2). An increase in P_n_ and R_a_ was advantageous because an increase water adsorption sites on membrane surfaces could potentially result in the improved water permeability of the modified PA–TFC membranes. Therefore, this phenomenon observed with the modified membranes could explain the increase in water permeability of the membranes with a higher R_a_ and P_n_. It can be inferred that an increase of R_a_ and P_n_ properties led to improved water molecule adsorption sites on membrane surfaces [50]. For example, the P_n_ obtained for the membranes modified with 1.00% AA, 0.50:1.00% ZnO/AA, 1.00:1.00% ZnO/AA, and 1.50:1.00% ZnO/AA were found to be 169, 170, 139, and 157, respectively. The unmodified membrane had a P_n_ of 141 (Table 2). The P_n_ values were higher than unmodified membranes, thereby leading to improved water permeability, as shown in Table 2. (The water permeability will be discussed in detail in Section 3.6 and Figure 7). 

The high distribution (H_d_) from the AFM for these membranes was used in this study to estimate the depth of the “troughs” found on the membrane surfaces. This is because it was assumed that the deeper the “troughs”, the more sites would be available for water and solute adsorption and entrapment, respectively [51]. Consequently, there was an increase in surface area, water permeability, and potentially increased membrane fouling. The H_d_ of the unmodified membrane was found to be 712.89 nm. The membranes modified with 1.00% ZnO/AA, 0.50:1.00% ZnO/AA, 1.00:1.00% ZnO/AA, and 1.50:1.00% ZnO/AA were found to possess H_d_ values of 693.04 nm, 850.92 nm, 355.86 nm, and 151.10 nm, respectively. On the other hand, the membranes modified with 1.50% ZnO/AA, 0.50:1.50% ZnO/AA, 1.00:1.50% ZnO/AA, and 1.50:1.50% ZnO/AA had a H_d_ of 250.83 nm, 727.14 nm, 151.10 nm, and 286.11 nm, respectively. From these results, it was observed that the membranes modified with ZnO/AA had their H_d_ values decrease with an increase in ZnO nanoparticle concentration. This phenomenon further confirmed that an increase of the ZnO nanoparticle concentration on the modified PA–TFC membranes with a constant AA concentration improved the smoothness of the membranes. Furthermore, the membranes modified with 1.00% AA and ZnO nanoparticles were much smoother than the membranes modified with 1.50% AA and ZnO nanoparticles according to the R_a_ values found in Table 2. This suggested that the optimum AA concentration was 1.00% AA and 0.5% ZnO nanoparticles, as this resulted in improved smoothness of the membranes (Figure 4 and Figure A1 (Appendix A)).

### 3.3. X-ray Diffraction Spectroscopic Analysis

The crystallinity and the presence of the ZnO nanoparticles on the PA–TFC membrane surfaces were determined using XRD spectroscopic analysis. Notably, the modified and unmodified PA–TFC membranes were amorphous, which was consistent with reports in the literature. However, there was progressively increasing crystallinity, even though the membranes remained predominantly amorphous. The progressively increasing crystallinity of the amorphous membrane material from XRD spectra was estimated using the intensity of the peaks and the shifting of the 2θ angles. It was clear from Figure 5a,b that the 2θ angles showed no shifting when the ZnO nanoparticles and/or AA hydrogels were incorporated into the PA–TFC membranes. However, the differences in peak intensities observed for the modified PA–TFC membranes in Figure 5a, consisting of the 1.00% AA concentration and increasing concentration of ZnO nanoparticles, showed higher intensity when compared to the membranes in Figure 5b (membranes modified with 1.50% AA, with increasing concentrations of ZnO nanoparticles). This observation could imply that the membranes modified with 0.50:1.00% ZnO/AA, 1.00:1.00% ZnO/AA, and 1.50:1.00% ZnO/AA, as shown in Figure 5a, even though they were amorphous in nature, possessed more crystalline properties than the membranes modified with 0.50:1.50% ZnO/AA, 1.00:1.50% ZnO/AA, and 1.50:1.50% ZnO/AA, as shown in Figure 5b. 

Previous research has shown that progressively increasing crystallinity plays an important role in influencing the flow rate of water molecules through the polyamide thin film matrix of the membranes. This is because the crystalline nature of the polyamide thin film surface can influence the water permeability of the PA–TFC membranes. Previous research has shown that polyamide thin films that possess crystalline matrices are water impermeable when compared to amorphous thin film matrices [52]. Consequently, crystalline membrane surfaces can lead to treacherous water pathways resulting in limited pathways available for water molecules to penetrate and diffuse through the polyamide thin films [53]. Conversely, according to Wagh et.al, amorphous membranes are known to have high water and solute diffusion when compared to crystalline membranes [54].

In Section 3.6, Figure 8a, we discussed in detail the enhanced water permeability of the modified membranes. Notably, the modified membranes (membranes modified with 1.00% AA and ZnO nanoparticles) were found to rather have higher permeability when compared to membranes modified with 1.5% AA and ZnO nanoparticles (Figure 8b). However, according to Figure 5, the same membranes with higher water permeability, even though they are predominantly amorphous, demonstrated increasing crystalline nature. Fundamentally, this contrasted with the previous research that demonstrated that higher crystallinity translated into reduced water permeability [52,53,54]. This could be explained by considering that other membrane surface properties, such as increasing roughness, could potentially promote the availability of water adsorption sites due to increased surface area. Furthermore, the modified membrane hydrophilicity contributed much more significantly to the water permeability performances than the increasing crystalline nature of the amorphous membranes. 

In addition, the ZnO nanoparticle characteristic peaks were clearly visible on the XRD spectral analysis of the membranes modified with 1.50:1.00% ZnO/AA, as shown in Figure 5a. This suggested a higher concentration of ZnO nanoparticles available on the surfaces of this particular modified membrane (1.00% AA) compared to other modified membranes (1.50% AA), as shown in Figure 5a. The limited appearance of the ZnO nanoparticle characteristic peaks for the 1.5% AA modified membranes, as shown in Figure 5b, was assigned to the higher concentration of AA (1.50%). The abundant presence of AA could potentially shield the detection of ZnO nanoparticles when compared to the slightly lower 1.00% AA concentration. This could therefore suggest that the interaction between the 1.50% AA and ZnO nanoparticles inside the polyamide thin film matrices could potentially suppress the detection of the ZnO characteristic peaks, even at higher concentrations of ZnO nanoparticles. Thus, the conclusion was that the ZnO nanoparticles were encapsulated into the AA matrices, making the nanoparticles more embedded into the polyamide thin film. This characteristic could further prevent leaching of the ZnO nanoparticles from the membranes. 

### 3.4. Thermal Gravimetric Analysis

Thermal gravimetric analysis (TGA) was used to investigate the thermal stability of the modified membranes [55]. The TGA curves of the unmodified PA–TFC membranes and membranes modified with 0.50:1.00% ZnO/AA, 1.00:1.00% ZnO/AA, and 1.50:1.00% ZnO/AA are shown in Figure 6a. The membranes modified with 0.50:1.50% ZnO/AA, 1.00:1.50% ZnO/AA, and 1.50:1.50% ZnO/AA are shown in Figure 6b. Table 3 shows the summary of the weight loss analysis and the thermal phases of the membranes.

The first phase of percentage weight loss for membranes was shown in Figure 6a between the temperatures of 169 °C and 394 °C. This weight loss of the membranes indicated the loss of moisture adsorbed onto the PA–TFC membranes [56]. The percentage weight loss at this stage was found to be 1.84% (unmodified membrane), 2.25% (1.00% AA), 4.26% (0.50:1.00% ZnO/AA), 2.94% (1.00:1.00% ZnO/AA), and 4.00% (1.00:1.50% ZnO) (Table 3). The second phase that showed the initial decomposition stage was found to range between 400 °C and 516 °C. This stage demonstrated a sharp percentage weight loss that was attributed to the acrylic acid hydrogels and the humidity adsorbed in the interior of ZnO nanoparticles [57]. The percentage weight loss observed for the unmodified and respective modified membranes shown in Figure 6a was found to be 82.20%, 78.89%, 72.20%, 74.03%, and 71.10%, respectively (Table 3). The third percentage weight loss event was found between 520 °C and 623 °C. This was attributed to the degradation of the polyamide backbone of the membranes [58]. The percentage weight loss of the respective membranes was found to be 7.51%, 6.83%, 8.06%, 15.48%, and 8.06% (Table 3). The membrane modified with 1.50:1.00% ZnO/AA showed a higher thermal stability when compared to the other membranes, as it showed a lower total percentage weight loss of 83.16% (Table 3). The total percentage weight loss of the respective membranes was found to be 91.55%, 87.97%, 84.52%, 92.45%, and 83.16%.

Figure 6b shows modified 1.50% AA and ZnO nanoparticles that demonstrated three phases of percentage weight loss as the temperature increased. The first phase showed the loss of moisture adsorbed on the membranes occurring between 220 °C and 375 °C [56]. The percentage weight loss obtained between 220 °C and 400 °C for the membranes modified with 1.50% AA, 0.50:1.50% ZnO/AA, 1.00:1.50% ZnO/AA, and 1.50:1.50% ZnO/AA in Figure 6b was found to be 1.84%, 2.12%, 2.05%, and 2.02%, respectively (Table 3). The second percentage weight loss phase occurred between 357 °C and 526 °C, as shown in Figure 6b. This was also attributed to the loss of acrylic acid hydrogels and moisture adsorbed in the interior of ZnO nanoparticles on the membranes [58]. The percentage weight loss observed of this phase for the respective membranes was found to be 76.61%, 74.42%, 74.30%, and 75. 28% (Figure 6b). This was a significant weight loss observed for the membranes as it was the biggest constituent of the membrane matrices.

The third phase represented the final degradation step ranging between 516 °C and 613 °C. The percentage weight loss was attributed to the degradation of the polyamide backbone [59]. The sample weight loss of the respective modified membranes in Figure 6b was found to be 8.07%, 10.14%, 6.84%, and 5.58%, respectively. The total percentage weight loss for the respective membranes was found to be 86.8%, 86.61%, 83.16%, and 84.56% (Figure 6b, Table 3). Even for this case, the membranes modified with 1.50:1.50% ZnO/AA showed a lower total percentage weight loss as compared to the other PA–TFC membranes. This could be because of the high percentage of the hydrogel and ZnO nanoparticles incorporated into the membranes.

### 3.5. Membrane Hydrophilicity Evaluation

Contact angle analysis was performed to evaluate the hydrophilicity of the membranes. Hydrophilicity is known to influence the water permeability of membranes [60]. The contact angle measurements of the unmodified PA–TFC membranes and modified membranes are presented in Figure 7a. The modified membranes displayed pH-responsiveness properties towards the contact angle measurements. This behavior was insignificant towards unmodified membranes. For the modified membranes, hydrophilicity was shown to increase with a decrease in pH level. For example, the membranes modified with 0.50:1.00% ZnO/AA, 1.00:1.00% ZnO/AA, and 1.50:1.00 ZnO/AA (Figure 7a) showed lower contact angles at pH 3 as compared to the other modified membranes in Figure 7b. 

Technically, the contact angles of the modified membranes decreased because of the ZnO nanoparticles and AA hydrogel indicating enhanced hydrophilicity of the modified membranes. However, the contact angles of the membranes in Figure 7a,b slightly increased when the concentration of the ZnO nanoparticles was increased. This was particularly observed with the membranes modified with 0.50% ZnO nanoparticles on both 1.00% and 1.50% AA modified membranes having the lowest contact angle. However, the concentration of the ZnO nanoparticles increased to 1.00% and 1.50%, the contact angles increased, most particularly at pH 3. This suggested that an increase in ZnO nanoparticle concentration led to a slight decrease in hydrophilicity, even though the membranes were generally more hydrophilic unmodified membranes. 

Furthermore, the introduction of the ZnO nanoparticles demonstrated an increase in hydrophilicity compared to membranes modified with AA hydrogels only. For example, the membranes modified with 1.00% AA, 0.50:1.00% ZnO/AA, 1.00:1.00% ZnO/AA, and 1.00:1.50% ZnO/AA had contact angles of 59.69 ± 4.82°, 45.31 ± 1.39°, 49.96 ± 5.11°, and 55.00 ± 0.12°, respectively, at pH 3. The unmodified membrane had the contact angle 80.69 ± 2.02° at pH 3. This clearly showed that incorporation of the AA hydrogels and ZnO nanoparticles improved the hydrophilicity on the modified membrane surfaces. This could be explained by the addition of carboxylic functional groups from the AA hydrogels and hydroxyl functional groups from the ZnO nanoparticles. These functional groups are known to improve preferential water adsorption on modified membrane surfaces [61].

Notably, the change in pH level increased the hydrophobicity of the modified membranes. As such, the increased contact angles of the modified membranes at pH 7 were rather higher than pH 3. For example, the membranes modified with 1.00% AA, 0.50:1.00% ZnO/AA, 1.00:1.00% ZnO/AA, and 1.00:1.50% ZnO/AA were found to have contact angles at 74.68 ± 1.06°, 59.23 ± 5.30°, 71.63 ± 6.44°, and 70.00 ± 1.40°. The unmodified membrane was found to be 101.78 ± 3.96°. Similarly, the contact angles for the respective modified membranes increased at pH 11 up to 89.50 ± 0.02°, 75.31 ± 1.80°, 78.51 ± 0.52°, and 83.00 ± 5.98°. At this pH, the unmodified membrane presented a contact angle of 112.82 ± 1.27°. A similar observation was made for the membranes modified with 1.50% AA, 0.50:1.50% ZnO/AA, 1.00:1.50% ZnO/AA, and 1.50:1.50% ZnO/AA, as shown in Figure 7b. These, respectively, had lower contact angles of 63.40 ± 3.71°, 48.06 ± 3.74°, 60.90 ± 2.50°, and 61.34 ± 1.77°, at pH 3. At pH 7, the respective membranes had contact angles of 69.45 ± 3.15°, 59.52 ± 2.90°, 54.61 ± 1.44°, and 61.93 ± 1.58°. There was, however, a sharp increase in contact angles at pH 11 of up to 78.12 ± 1.53°, 67.72 ± 4.52°, 89.90 ± 3.03°, and 80.98 ± 0.54°, respectively. This contact angle behavior clearly demonstrated the pH responsiveness of the modified membranes with AA hydrogels.

Generally, it was observed that the modified membranes in Figure 7b had higher contact angles when compared to the modified membranes in Figure 7a. This implied that the membranes modified with 1.00% AA hydrogels and increasing ZnO nanoparticle concentrations were more hydrophilic when compared to the membranes modified with 1.50% AA, with increasing ZnO nanoparticles concentrations. All the membranes showed lower contact angles at pH 3 and increased as the pH increased to pH 7 and 11. This clearly confirmed the “gate effect” properties that these modified membranes possessed, imparted by the AA hydrogel to the membrane. Additionally, at lower pH levels, the carboxylic and hydroxyl functional groups from the respective AA and ZnO nanoparticles were protonated as the hydrogel polymers shrank. This in turn, allowed for more water molecules to adsorb onto the membrane surfaces. However, under basic pH conditions, the acrylic acid’s carboxylic acid was deprotonated into –COO^–^ ions [53]. Consequently, this caused the hydrogel to swell and presented preferential adsorption towards dissolved solute ions rather than water molecules. Thus, decreased attachment of water molecules was experienced, and this could explain the increased hydrophobic nature of membranes at higher pH levels (pH 7 and 11). Furthermore, the decrease in hydrophilicity of the modified membranes as a result of the increase in ZnO nanoparticle concentrations could be due to the non-uniform distribution of ZnO nanoparticles as observed in SEM images [20].

### 3.6. Water Permeability of the Membranes

Water permeability analysis of the membranes as a function of pH was performed to evaluate the effect of the incorporation of AA and ZnO nanoparticles onto the polyamide thin films of the PA–TFC membranes (Figure 8). The PA–TFC membranes modified with 1.00% AA, 0.50:1.00% ZnO/AA, 1.00:1.00% ZnO/AA, and 1.50:1.00% ZnO/AA demonstrated water permeability of 2.83 ± 0.65 L·m^−2^·h^−1^·kPa^−1^, 9.43 ± 0.60 L·m^−2^·h^−1^·kPa^−1^, 7.67 ± 0.32 L·m^−2^·h^−1^·kPa^−1^, and 6.06 ± 0.18 L·m^−2^·h^−1^·kPa^−1^, respectively, at pH 3 (Figure 8a). This was found to be greater when compared to water permeability at pH 7 and 11 levels. It was also observed that the membrane modified with 0.50:1.00% ZnO/AA had the highest water permeation (9.43 ± 0.60 L·m^−2^·h^−1^·kPa^−1^ ) at pH 3. This observation corresponded to its superior hydrophilic nature and the increased roughness observed. The unmodified membrane had a water permeability of 2.29 ± 0.59 L·m^−2^·h^−1^·kPa^−1^ at pH 3. This was found to be slightly lower than that of the water permeability for the membrane modified with 1.00% AA (2.83 ± 0.65 L·m^−2^·h^−1^·kPa^−1^ ). This could suggest that the incorporation of the ZnO nanoparticles further improved the hydrophilic nature of the modified membranes. Consequently, it was clearly inferred that the addition of ZnO nanoparticles further improved water permeability at lower pH levels than the membranes modified with 1.00% AA hydrogels only (Figure 8)

Water permeability of the membranes modified with 1.00% AA, 0.50:1.00% ZnO/AA, 1.00:1.00% ZnO/AA, and 1.50:1.00% ZnO/AA showed a decrease at pH 7. The respective membranes had water permeabilities of 2.65 ± 0.08 L·m^−2^·h^−1^·kPa^−1^, 5.34 ± 0.42 L·m^−2^·h^−1^·kPa^−1^, 3.02 ± 0.36 L·m^−2^·h^−1^·kPa^−1^ and 2.85 ± 0.91 L·m^−2^·h^−1^·kPa^−1^ (Figure 8b). The unmodified membranes had a lower water permeability of 1.79 ± 0.27 L·m^−2^·h^−1^·kPa^−1^. A further decrease was observed for the respective membranes at pH 11, where their water permeabilities were found to be 0.9 ± 0.21 L·m^−2^·h^−1^·kPa^−1^, 1.80 ± 0.22 L·m^−2^·h^−1^·kPa^−1^, 1.83 ± 0.52 L·m^−2^·h^−1^·kPa^−1^, 0.80 ± 0.41 L·m^−2^·h^−1^·kPa^−1^ and 0.83 ± 0.25 L·m^−2^·h^−1^·kPa^−1^. The unmodified membranes had an even lower water permeability of 0.90 ± 0.21 L·m^−2^·h^−1^·kPa^−1^. This phenomenon clearly proves pH responsiveness of the membranes due to the presence of acrylic acid hydrogels in the polyamide thin film.

The water permeability of the membranes modified with 1.00% AA and ZnO nanoparticles (Figure 8a) was found to be slightly higher than the membranes modified with 1.50% AA and ZnO nanoparticles (Figure 8b). However, there was a significant increase of the latter modified membranes when compared to the unmodified membranes. Moreover, the membranes modified with 1.50% AA and ZnO nanoparticles showed pH responsiveness across all the pH levels, apart, however, from the membranes modified with 0.5:1.50% ZnO/AA, which showed a slight increase of water permeability at pH 7 but a drop at pH 11. Generally, the membranes modified with 1.00% AA and ZnO nanoparticles showed higher pH responsiveness when compared to the membranes modified with 1.50% AA. This was because at a lower pH, the AA brushes collapsed, leading to a higher passage of water molecules. This contrasted with higher pH levels, such as the pH 7 and 11 used in this research work, where the brushes tend to swell, thereby restricting the passage of water molecules through the membrane matrices.

### 3.7. Membrane Swelling Evaluation

The swelling degree of the membranes was studied since swelling of the PA–TFC membranes can influence water permeability [62]. This is because higher swelling has been shown to result in membranes with lower water permeability, as water molecules struggle to diffuse through membrane matrices due to the swollen barrier [62]. The swelling degree percentage as a function of pH of the PA–-TFC membranes was shown in Figure 9a,b. It was observed that the swelling degree increased with an increase in pH level. This was because at a pH level higher than the pKa of 4.74 (which is the pKa of the acrylic acid hydrogel), the AA brushes deprotonate, leading to increased swelling of the membrane. This could lead to the restriction of the water molecules’ passage through the modified membrane [63]. This phenomenon explains the lower water permeability of the modified membranes at pH 11, as shown in Figure 8. Contrary to higher pH levels, at lower pH levels the AA brushes collapsed due to the protonation of COO^–^ ion into a –COOH group. Hence, there was less swelling observed at pH 3 when compared to pH 7 and 11 [63]. Furthermore, the collapse of the AA hydrogels reduced the steric hindrance against water passage. Hence, the high-water permeability of the membranes at pH 3. 

The unmodified membranes demonstrated a percentage swelling degree of 66.67 ± 0.8%. The modified membranes showed percentage swelling degrees of 48.62 ± 0.21% (1.00% AA), 7.75 ± 1.03% (0.50:1.00% ZnO/AA), 26.08 ± 0.61% (1.00:1.00% ZnO/AA), and 14.00 ± 0.06% (1.50:1.00% ZnO/AA) at pH 3. From this low swelling degree, the percentage swelling degree increased to 92.78 ± 0.33% (unmodified membrane), 72.79 ± 0.01% (1.00% AA), 78.39 ± 0.36% (0.50:1.00% ZnO/AA), 61.01 ± 0.37% (1.00:1.00% ZnO/AA), and 67.70 ± 2.60% (1.50:1.00% ZnO/AA) at pH 7 (Figure 8). This clearly showed that the modified membranes were pH responsive as the membranes increased in percentage swelling with increasing pH levels. As such, the membranes were also slightly responsive to the changes in pH but were not as significantly responsive as the modified membranes. 

Notably, the water permeability trends of the unmodified membranes across the different pH levels were lower than the modified membranes and passively followed the trends of the modified membranes. This observation suggested that the slight pH responsiveness of the unmodified membranes was not due to their sensitivity to the varying pH conditions resulting into the “gate effect” phenomenon. Rather, this passive trend was due surfaces charges because of the varied pH conditions. Furthermore, the modified membranes showed lower percentage swelling degrees than the unmodified membranes. This means that the AA and ZnO nanoparticles decreased the swelling degree of the membranes. This property could potentially result in a membrane with superior water permeability properties, as observed with these modified membranes. 

### 3.8. Pb(II) Rejection Analysis

The rejection of Pb (II) ions as a function of pH was performed with the differently modified membranes, as shown in Figure 10. To explain the rejection of the membranes, particular emphasis was focused on the swelling phenomenon, gating effect, and ion exchange capacity of the membranes, as these properties demonstrated great influence on the membrane’s performance behavior [63,64]. Highly swollen membranes due to the AA hydrogel brushes demonstrating gating effect characteristics are known to demonstrate higher rejection efficiencies [62]. Based on Figure 10, the Pb (II) ion rejection improved with an increase in pH level. This contrasted with the rejection of the unmodified membranes, which showed a decline with an increase in pH levels. The unmodified membrane at both pH 3 and 7 showed higher rejection performances than modified membranes. However, at pH 11, the rejection was lower than for unmodified membranes (Figure 10). 

This rejection pattern was interesting because the water permeability (Figure 8) was almost following the trend of the modified membranes, even though the unmodified membrane permeability was lower for the unmodified membranes. This clearly showed that the rejection mechanism of the unmodified membranes could not be attributed to the “gate effect” because of pH responsiveness. It should be noted that, at pH 11, the percentage swelling degree of modified and unmodified membranes was similar (Figure 9). Nonetheless, with similar percentage swelling degrees for both modified and unmodified membranes, the rejection behavior remained different. This was because the rejection trend of unmodified membranes remained different to that of modified membranes, confirming that the “gate effect” was not influencing the rejection behavior of the membranes. This is because the unmodified membranes were not following the “gate effect” due to pH responsiveness, as were the AA modified membranes. Therefore, the rejection trend of unmodified membranes was different from that of the modified membranes as demonstrated in Figure 10. 

The enhanced rejection capacity of the modified membranes at pH 7 and 11 can be explained by the “gate effect” phenomenon. As previously explained, at basic pH, the AA hydrogel brushes have the tendency of expanding due to the abundant –OH- anions from the adjusted pH. Consequently, the expansion of the AA hydrogel brushes within the polyamide thin film effectively restricts the passage of water molecules and dissolved solutes as shown in Figure 8, Figure 9 and Figure 10. Furthermore, at this pH level, the –COOH functional groups of the AA hydrogels had depronated into –COO- anions. This could, therefore, lead to the adsorption of the Pb (II) heavy metal ions to the side chains. On the other hand, at lower pH levels, the Pb (II) ion rejection was lower because the hydrogels had collapsed. Consequently, that phenomenon led to their easier passage through the polyamide thin film matrices. Additionally, the –COOH side chains of the AA had protonated, thus desorbed the Pb (II) heavy metal ions. For example, the membranes modified membranes with 1.00% AA, 0.50:1.00% ZnO/AA, 1.00:1.00% ZnO/AA, and 1.50:1.00% ZnO/AA had rejections of 10.36 ± 1.30%, 19.84 ± 0.21%, 29.85 ± 0.06%, and 7.03 ± 0.08%, respectively, at pH 3 (Figure 10a).

The unmodified membranes showed 99.14 ± 2.90% rejection of Pb (II) ions at pH 7. The modified membranes were found to be 25.92 ± 2.07%, 17.76 ± 3.28%, 14.98 ± 0.04%, and 20.03 ± 0.05%, irrespectively. The Pb (II) ion rejection of the unmodified membrane at pH 11 was found to be 80.63 ± 4.00%, showing a decrease in rejection. However, membranes modified with different weigh percentages of ZnO/AA showed high Pb (II) ion rejection at pH 11: 84.36 ± 0.01%, 92.90 ± 0.28%, 96.68 ± 2.57%, and 98.58 ± 0.35%, respectively.

The trend observed in this study clearly proved the “gate effect” these membranes possessed due to their pH responsiveness and the protonation–deprotonation effect of the –COOH functional groups. The Pb (II) ion rejection of the unmodified membrane at pH 11 was shown to be 49.75 ± 1.00%. This was a low rejection of the Pb (II) ions, demonstrating that even though these unmodified membranes were slightly responding to pH, it was not caused by the “gate effect”, but simply due to changes in the pH conditions of the feed water. This phenomenon was explained by the fact that the percentage swelling degree of all the modified and unmodified membranes was similar at pH 11, while the rejection performances were inversely proportional. However, the behavior of the unmodified membranes does not seem to follow the trend of the modified membranes. This was due to the limited pH responsiveness afforded by the AA hydrogels. 

The membranes modified with 1.50% AA, 0.50:1.50% ZnO/AA, 1.00:1.50% ZnO/AA, and 1.50:1.50% ZnO/AA showed a Pb (II) ion rejection of 27.93 ± 0.11%, 16.11 ± 0.05%, 47.20 ± 2.20%, and 24.84 ± 2.24%, respectively, at pH 3. At pH 7, the rejection obtained for the respective modified membranes was 16.08 ± 0.33%, 30.58 ± 0.13%, 24.67 ± 0.02%, and 33.90 ± 0.31%. It was observed that the rejection also increased sharply as the pH increased to pH 11, similarly to the membranes modified with 1.00% AA and ZnO nanoparticles. This is because the Pb (II) ion rejection obtained for the respective modified membranes was found to be 91.98 ± 0.09%, 96.67 ± 2.00%, 90.85 ± 1.19%, and 95.55 ± 0.01%. This was also clear proof of the “gate effect” property because of the incorporation of AA hydrogels in the polyamide thin films. Moreover, the membranes modified with 0.50:1.50% ZnO/AA gave the highest Pb (II) ion rejection when compared to the other modified membranes (Figure 10b). However, it was observed this was lower than the membrane modified with 0.50:1.00% ZnO/AA (Figure 10a). These superior rejection efficiencies were thought to be due to the optimum 0.50% concentration of ZnO nanoparticles, as had been previously reported [21,52].

### 3.9. Ion Exchange Capacity Analysis

Ion exchange capacity (IEC) analysis was performed in order to determine the charge density on the membrane surface which would assist in determining the pH responsiveness of the PA–TFC membranes with respect to their rejection behavior [64,65]. Figure 11 showed the IEC of the PA–TFC membranes with different concentrations of ZnO nanoparticles, constant AA concentration, and Pb(II) ions rejection as a function of pH. The IEC of the unmodified membrane was found to be 40.82 ± 1.27 mmol/g, whilst 61.89 ± 1.89 mmol/g and 54.99 ± 0.856 mmol/g was observed for the membranes modified with 1.00% AA and 1.50% AA, respectively (Figure 11a). It was observed that both modified PA–TFC membranes had a higher IEC when compared to unmodified membranes. Additionally, the membrane modified with 1.00% AA was found to possess the highest IEC when compared to membranes modified with 1.50% AA. The increase in IEC was a result of AA concentration increase, as there were more binding sites for ion exchange (–COOH functional groups). Moreover, this was advantageous in the membrane matrices, as it could result in higher probabilities of improved rejection capacity of heavy metals because of increased ionic binding sites on the modified membranes surfaces.

Rejection capacity at pH 3, followed by pH 7 was found to be higher than that at pH 11 for the unmodified membranes (Figure 10). However, the rejection capacity of the modified membranes significantly decreased at pH 3 and 7 compared to unmodified membranes, as shown in Figure 11a. This occurred despite the increased IEC of the modified membranes and clearly demonstrated the effect of the incorporation of the AA hydrogels into the membranes, signifying the “gate effect” mechanism possessed by the modified membranes, irrespective of binding sites that the IEC demonstrated to have increased. Therefore, there was an inverse relationship between the rejection capacity at pH 3 and 7 with the IEC, with an AA hydrogel concentration increment. In this case, as the AA hydrogel was increased, the rejection capacity decreased, even though the IEC had increased. On the other hand, different rejection behavior was observed for the same membranes at pH 11, namely, a direct proportional relationship between rejection capacity, IEC, and AA concentration increments. At pH 11, it was observed that the IEC increased with an increase in rejection capacity, as the concentration of the AA hydrogel increased (Figure 11a). In addition, for this particular pH level, when the IEC was low, the rejection capacity of the Pb (II) ions was also lower. Consequently, this clearly demonstrated that the modified membranes were swollen at higher pH conditions than lower pH conditions, where the hydrogel polymers would be shrunk.

Figure 11b showed the IEC behavior and rejection capacity of the membranes modified with 1.00% AA and increasing concentration of ZnO nanoparticles. The IEC for the membranes modified with 0.50:1.00% ZnO/AA, 1.00:1.00% ZnO/AA, and 1.50:1.00% ZnO/AA was found to be 55.89 ± 2.62 mmol/g, 29.79 ± 0.89 mmol/g, and 21.64 ± 1.05 mmol/g, respectively. After the membrane modified with 1.00% AA at 61.89 ± 1.89 mmol/g, the membrane modified with 0.50:1.00% ZnO/AA had a higher IEC when compared to the membranes modified with 1.00:1.00% ZnO/AA and 1.50:1.00 ZnO/AA. It was also noticed that the IEC decreased drastically as the ZnO nanoparticle concentration increased in the modified membranes. However, it was observed in Figure 11b that the rejection capacity at pH 11 remained higher for all membrane types compared to the corresponding lower pH levels. A similar trend was observed for membranes modified with 1.50% AA and increasing ZnO nanoparticles. This also showed no relationship between IEC and rejection capacity, proving that the “gate effect” had more effect than IEC in the rejection behavior of the modified membranes.

It has already been stated that the SEM images showed an even distribution of the 0.50% ZnO nanoparticles when compared to higher ZnO nanoparticle concentrations. This observation was significant with respect to the IEC values of the modified membranes. Fox example, Figure 11c showed the membranes modified with 0.50:1.50% ZnO/AA, 1.00:1.50% ZnO/AA, and 1.50:1.50% ZnO/AA, with corresponding IEC values of 58.89 ± 0.99 mmol/g, 79 ± 1.17 mmol/g, and 17.64 ± 0.65 mmol/g, respectively. In this case, the membranes with an even distribution of ZnO nanoparticles, which were the membranes modified with 0.50:1.50% ZnO/AA, showed the highest IEC value when compared to the other modified membranes. The IEC of the membranes in Figure 11c was generally lower when compared to the membranes shown in Figure 11b. This could be explained by the high acrylic acid content (1.50% AA), which demonstrated a significant negative effect on the performance of the membranes, as observed with water permeability and rejection. However, the IEC value in this case corresponded with the Pb II) rejection at lower pH levels. Additionally, the Pb s(II) rejection increased with pH conditions because of the ZnO nanoparticles incorporated into the surface of the modified PA–TFC membranes.

## 4. Conclusions

Acrylic acid hydrogels and ZnO nanoparticles were successfully embedded onto polyamide thin film composite membranes through an in situ interfacial polymerization process. These modified membranes showed improved water permeability and Pb (II) heavy metal ions rejection when compared to unmodified membranes. It was also noted that the membranes modified with 0.50:1.00% ZnO and 0.50:1.50% ZnO/AA demonstrated higher water permeability and rejection performances. Additionally, the surface morphology showed that the membranes with an even distribution of the ZnO nanoparticles demonstrated better water permeability and rejection performance. The thermal stability of the membranes improved because of the incorporation of ZnO/AA dopants. Consequently, the membranes modified with 1.50:1.00% ZnO/AA and 1.50:1.50% ZnO/AA showed a higher thermal stability than the other membranes. It can be concluded that an increase in ZnO nanoparticles significantly improved the thermal stability of the modified membranes compared to unmodified membranes. Moreover, the incorporation of the AA hydrogels into the polyamide thin film composite membranes clearly demonstrated the presence of the “gate effect” mechanism which had been imparted to the modified membranes. This was due to the improved hydrophilicity, ion exchange capacity, and responsive swelling of membranes, resulting from the incorporated AA hydrogels and ZnO nanoparticles.

## Figures and Tables

**Figure 1 membranes-11-00910-f001:**
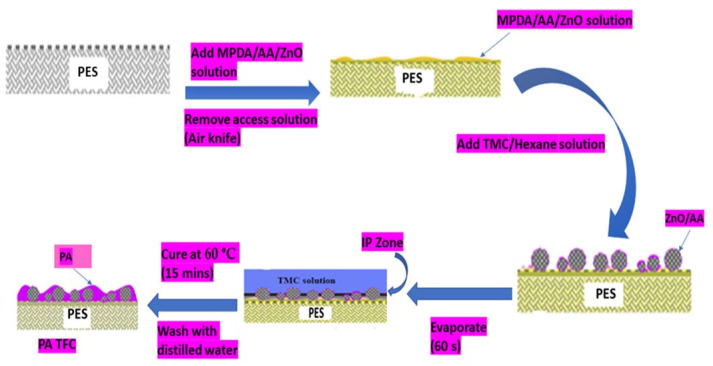
Schematic modification of PA TFC/ZnO/AA membranes through the Interfacial Polymerization (IP) method.

**Figure 2 membranes-11-00910-f002:**
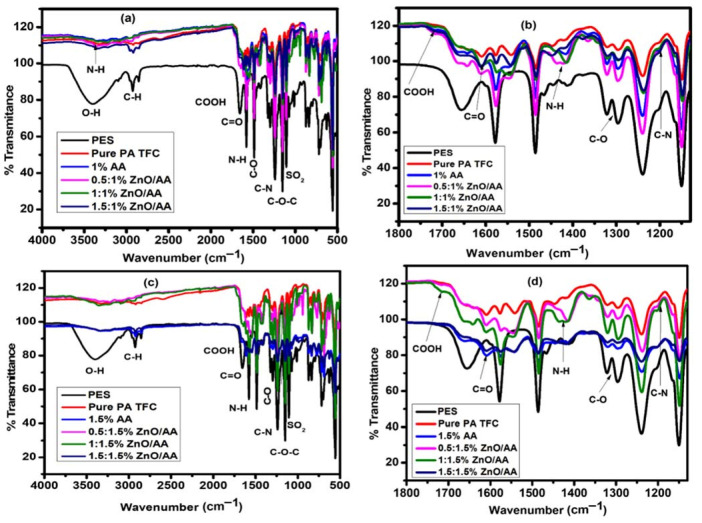
(**a**) ATR–FTIR spectra showing membranes modified with 0.00% AA (unmodified PA–TFC membrane), 1.00% AA, 0.50:1.00% ZnO/AA, 1.00:1.00% ZnO/AA, and 1.50:1.00% ZnO/AA. (**b**) Expanded spectra of (a) between 1800–1130 cm^−1^. (**c**) Membranes modified with 1.50% AA, 0.50:1.50% ZnO/AA, 1.00:1.50% ZnO/AA, and 1.50:1.50% ZnO/AA. (**d**) Expanded spectra of (**c**) between 1800–1130 cm^−1^.

**Figure 3 membranes-11-00910-f003:**
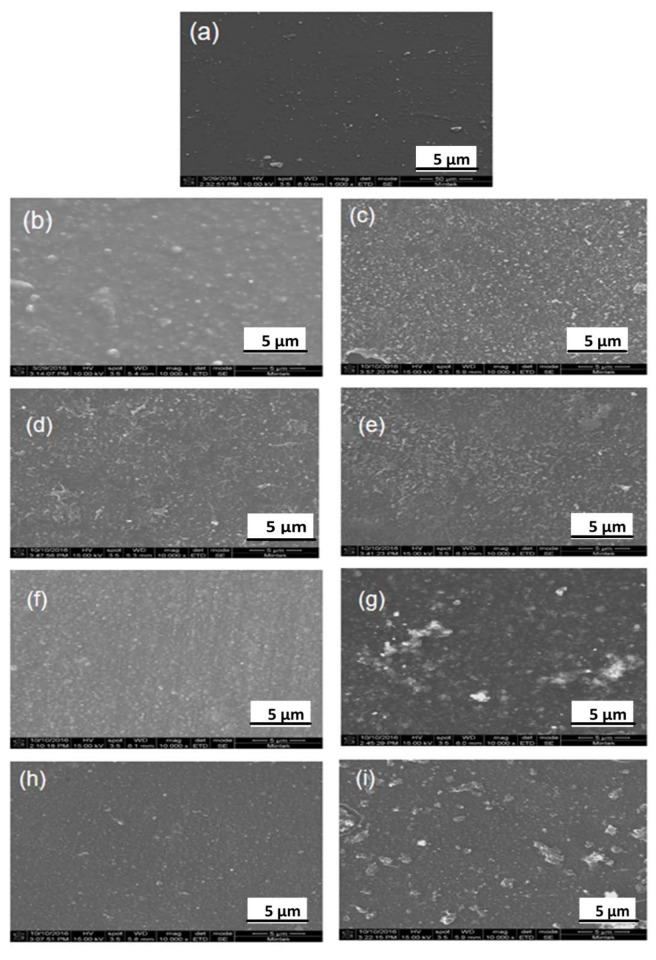
SEM images of: (**a**) unmodified membranes; and membranes modified with: (**b**) 1.00% AA, (**c**) 0.50:1.00% ZnO/AA, (**d**) 1.00:1.00% ZnO/AA, (**e**) 1.50:1.00% ZnO/AA, (**f**) 1.50% AA, (**g**) 0.50:1.50% ZnO/AA, (**h**) 1.00:1.50% ZnO/AA, and (**i**) 1.50:1.50% ZnO/AA.

**Figure 4 membranes-11-00910-f004:**
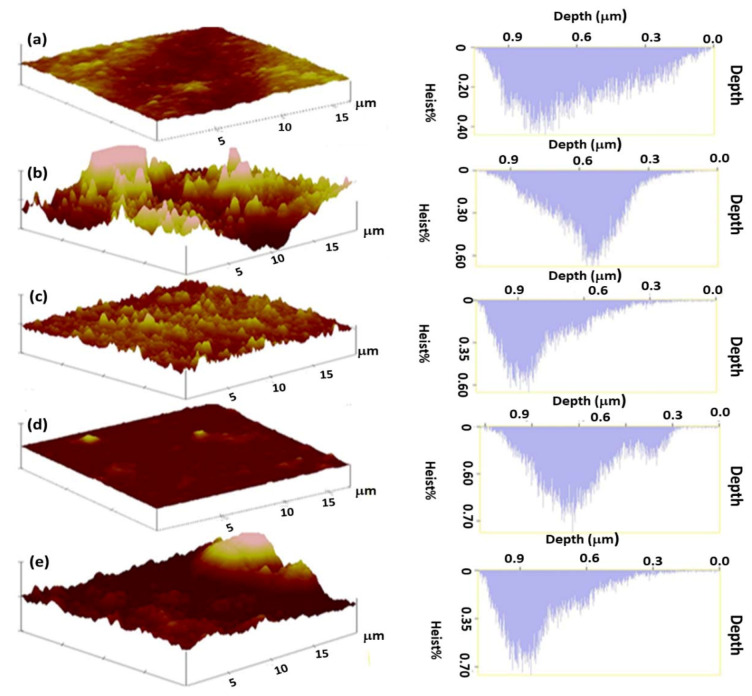
AFM images of: (**a**) unmodified PA–TFC membrane; and membranes modified with: (**b**) 1.00% AA, (**c**) 0.50:1.00% ZnO/AA, (**d**) 1.50% AA, and (**e**) 0.50:1.50% ZnO/AA.

**Figure 5 membranes-11-00910-f005:**
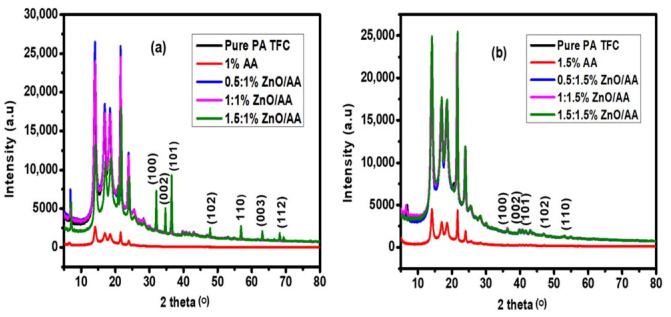
XRD spectra of: (**a**) unmodified membranes (0.00% ZnO/AA); and membranes modified by: 1.00% AA, 0.50:1.00% ZnO/AA, 1.00:100% ZnO/AA, and 1.50:1.00% ZnO/AA; and (**b**) membranes modified with 0.00% ZnO/AA, 1.50% AA, 0.50:1.50% ZnO/AA, 1.00:1.50% ZnO/AA, and 1.50:1.50% ZnO/AA.

**Figure 6 membranes-11-00910-f006:**
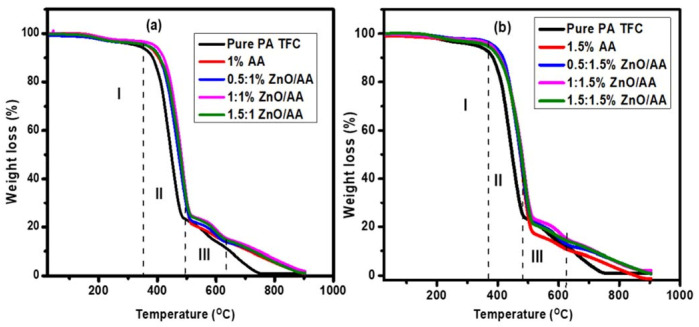
Thermal Gravimetric Analysis of: (**a**) unmodified membrane, and membranes modified with 1.00% AA, 0.50:1.00% ZnO/AA, 1.00:1.00% ZnO/AA, and 1.50:1.00% ZnO/AA; and: (**b**) membranes modified with 1.50% AA, 0.50:1.50% ZnO/AA, 1.00:1.50% ZnO/AA, and 1.50:1.50% ZnO/AA.

**Figure 7 membranes-11-00910-f007:**
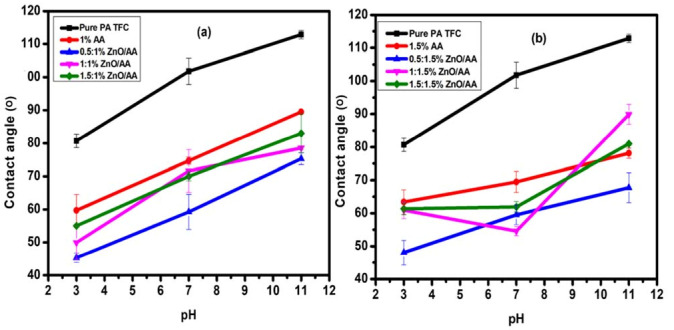
Contact angle analysis of: (**a**) membranes modified with 0.00% ZnO/AA, 1.00% AA, 0.50:1.00% ZnO/AA, 1.00:1.00% ZnO/AA, and 1.50:1.00% ZnO/AA; and: (**b**) membranes modified with 1.50% AA, 0.50:1.50% ZnO/AA, 1.00:1.50% ZnO/AA, and 1.50:1.50% ZnO/AA.

**Figure 8 membranes-11-00910-f008:**
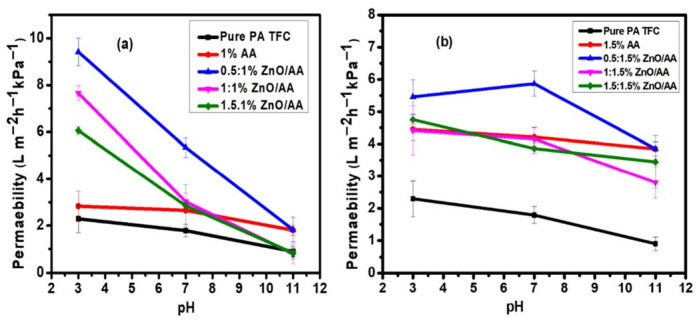
Water permeability of: (**a**) membranes modified with 0.00% AA, 1.00% AA, 0.50:1.00% ZnO/AA, 1.00:1.00% ZnO/AA, and 1.50:1.00% ZnO/AA; and: (**b**) membranes modified with 1.50% AA, 0.50:1.50% ZnO/AA, 1.00:1.50% ZnO/AA, and 1.50:1.50% ZnO/AA.

**Figure 9 membranes-11-00910-f009:**
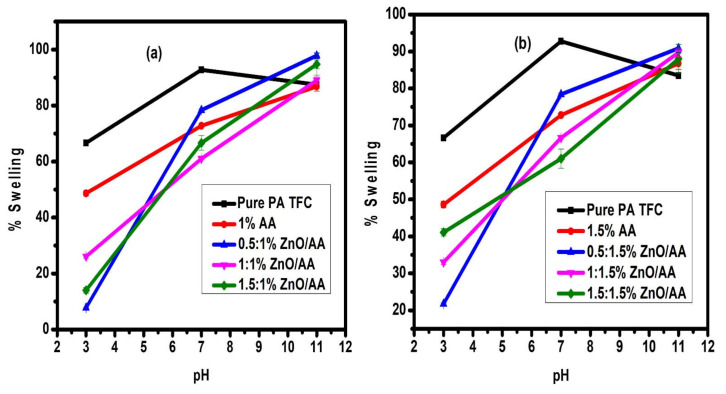
The percentage swelling degree analysis of: (**a**) membranes modified with 0.00% AA, 1.00% AA, 0.50:1.00% ZnO/AA, 1.00:1.00% ZnO/AA, and 1.50:1.00% ZnO/AA; and: (**b**) membranes modifies with 1.50% AA, 0.50:1.50% ZnO/AA, 1.00:1.50% ZnO/AA, and 1.50:1.50% ZnO/AA.

**Figure 10 membranes-11-00910-f010:**
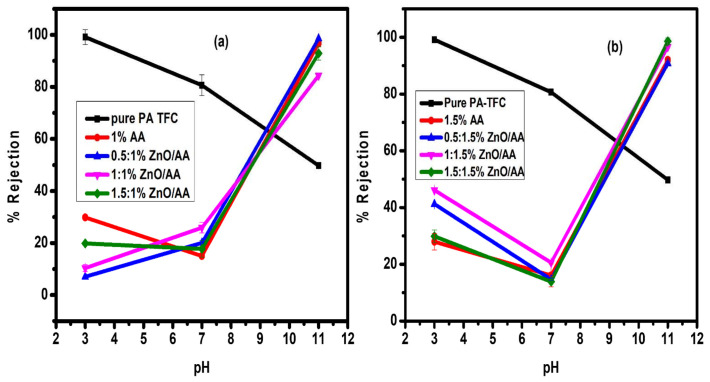
The Pb (II) ions rejection (%) of: (**a**) membranes modified with 0.00% AA, 1.00% AA, 0.50:1.00% ZnO/AA, 1.00:1.00% ZnO/AA, and 1.50:1.00% ZnO/AA; and: (**b**) membranes modifies with 1.50% AA, 0.50:1.50% ZnO/AA, 1.00:1.50% ZnO/AA, and 1.50:1.50% ZnO/AA as a function of pH.

**Figure 11 membranes-11-00910-f011:**
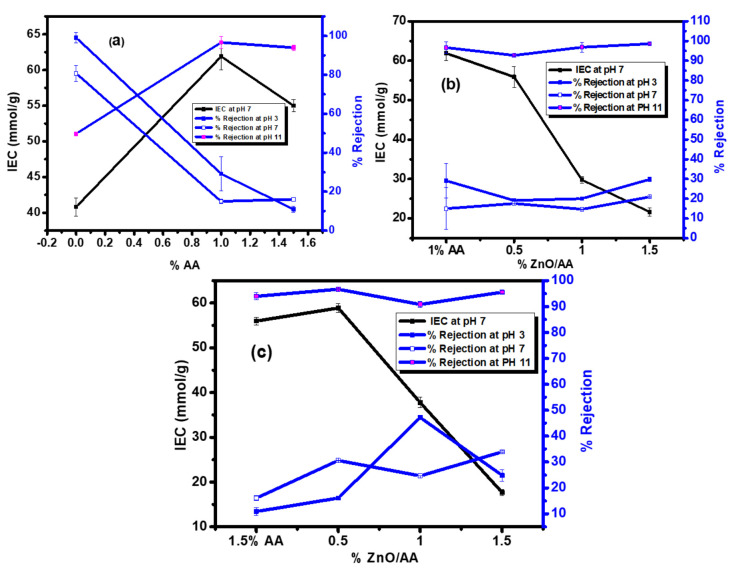
The variation of IEC performed at pH 7 for the PA–TFC membranes modified with 0.00% AA (unmodified membrane) and membranes modified with: (**a**) 1.00% AA, 1.5% AA; (**b**) 1.00% AA, 0.50:1.00% ZnO/AA, 1.00:1.00% ZnO/AA, and 1.50:1.00% ZnO/AA; and (**c**) 1.50% AA, 0.50:1.50% ZnO/AA, 1.00:1.50% ZnO/AA, and 1.50:1.50% ZnO/AA with respect to rejection capacity and pH conditions.

**Table 1 membranes-11-00910-t001:** Table showing different concentrations of m-phenylenediamine (MPDA) and acrylic acid (AA) on polyamide membrane surfaces.

MPDA (% *m/v*)	AA (% *m/v*)	ZnO (% *m/v*)
2.00	0.00	0.00
1.00	1.00	0.50
1.00	1.00	1.00
1.00	1.00	1.50
0.50	1.50	0.50
0.50	1.50	1.00
0.50	1.50	1.50

**Table 2 membranes-11-00910-t002:** Peak to peak maximum distance (S_y_), height distribution (H_d_), mean roughness (R_a_), root mean square roughness (R_q_), and peak densities (P_n_) obtained from AFM analysis (image scan area; 20 × 20 µm).

Membrane Type	R_q_ (nm)	R_a_ (nm)	S_y_ (nm)	H_d_ (nm)	P_n_
Unmodified Membranes	218.94	183.11	26.00	712.89	141
1.00% AA	11.85	77.63	17.59	693.04	169
0.50:1.00% ZnO/AA	354.25	218.80	38.28	850.92	170
1.00:1.00% ZnOAA	77.59	56.98	4.85	355.85	139
1.50:1.00% ZnO/AA	84.93	60.19	5.31	151.10	157
1.50% AA	62.85	50.13	4.02	250.83	150
0.50:1.50% ZnO/AA	111.85	208.80	2.18	727.14	163
1.00:1.50% ZnO/AA	193.40	146.56	30.03	151.10	152
1.50:1.50% ZnO/AA	71.72	56.23	7.43	286.11	156

**Table 3 membranes-11-00910-t003:** Summary of the percentage weight loss phases and decomposition stages of the membranes.

Names	Initial Stage		Degradation Stage				Total wt. Loss (%)
ZnO/AA Concentration (%)	Phase I		Phase II		Phase III		
	Initial Temp. (°C)	%wt. Loss	1st Deg Temp (°C)	%wt. Loss	2nd Deg Temp (°C)	%wt. Loss	
0.00	223–382	1.84	505–382	82.20	494–580	7.51	91.55
1.00 AA	224–381	2.25	381–511	78.89	516–642	6.83	87.97
0.50:1.00	260–400	4.26	400–510	72.20	520–612	8.06	84.52
1.00:1.00	244–387	2.94	387–518	74.03	522–619	15.48	92.45
1.50:1.00	246–392	4.00	392–516	71.10	522–623	8.06	83.16
1.50 AA	225–382	2.12	382–515	76.61	515–624	8.07	86.8
0.50:1.50	234–357	2.05	357–512	74.42	516–631	10.14	86.61
1.00:1.50	220–385	2.02	385–518	74.30	520–625	6.84	83.16
1.50:1.50	248–379	0.70	379–525	75.28	520–613	5.58	81.56

## Data Availability

The raw data will be made available when requested.

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
