# Peer review of "An In Situ Incorporation of Acrylic Acid and ZnO Nanoparticles into Polyamide Thin Film Composite Membranes for Their Effect on Membrane pH Responsive Behavior"

_membranes, 2021, doi:10.3390/membranes11120910_

Round 1

Reviewer 1 Report

  1. Table 1, percent ratio (%) of ZnO should be clarified. For example, what is the meaning of ‘0.50;1.00;1.50’?. Also, the percentages of chemicals used in this study should be clearly indicated whether those are calculated in the base of weight percent or volume percent.
  2. Page 5, line 210, mentioned that two optimum concentrations of ZnO/AA (1% and 1.5%). How are those optimum concentrations chosen?
  3. Please provides a schematic image to illustrate the incorporation mechanism of ZnO/AA into PA layer. Also, if those additives are physically incorporated in the PA layer and how about their stability?
  4. Figure 2, scale bars and magnifications of SEM images should be inserted.
  5. I suggest that the authors improve the quality of Figures 1, 4, 5, 6, 7, 8, 9 and 10, to clearly distinguish trends of each sample not only by different colours but also by different symbols and patterns.
  6. Page 10, lines 317-329, Figure 4 and 5 indicate results of XRD and TGA analysis for the samples. It does not represent hydrophilicity and permeability improvements by adding additives. Thus, these statements should be properly revised.
  7. Page 12, lines 393-395, again, Figure 4 does not represent permeability, so this should be revised.
  8. Page 19, lines 616-619, there is a redundancy of ‘unmodified membranes’ should be revised.

Author Response

Reviewer 1

Comments and Suggestions for Authors

  1. Table 1, percent ratio (%) of ZnO should be clarified. For example, what is the meaning of ‘0.50;1.00;1.50’?. Also, the percentages of chemicals used in this study should be clearly indicated whether those are calculated in the base of weight percent or volume percent.

Answer: Table 1, percentage has been addressed into percentage mass per volume (%m/v). The 0.5/1.00/1.50 were the different concentrations of ZnO nanoparticles. i.e., 1% MPDA, 1 AA, and 0.5% ZnO nanoparticles or 1.00% ZnO nanoparticles or 1.50% nanoparticles. This has been reflected in the manuscript.

  1. Page 5, line 210, mentioned that two optimum concentrations of ZnO/AA (1% and 1.5%). How are those optimum concentrations chosen?

Answer: This sentence was revised address to indicate the following:

“...For the membranes modified AA, two concentrations of 1.00% and 1.50% AA were chosen as constant concentrations. This means that at 1% m/v AA or 1.5% m/v AA concentration, the concentration of ZnO nanoparticles were gradually increased to study the impact of the ZnO nanoparticles. As such, for these AA concentrations, increasing concentrations of the ZnO nanoparticles were introduced and studied to investigate the effect of the increasing nanoparticle concentration as shown in Table 1. These concentrations of AA were determined after a series of optimization of AA concentration on the membranes using rejection capacity and water permeation as parameters for determining the appropriate loading concentration.”

  1. Please provides a schematic image to illustrate the incorporation mechanism of ZnO/AA into PA layer. Also, if those additives are physically incorporated in the PA layer and how about their stability?

Answer: The schematic imaged incorporated as shown Figure 1. The additives are indeed physically incorporated into the PA layer. However, they are stable into the PA thin film. This is because, based on the chemistry of acrylic acid, it would be chemically bound to the thin film because of the chemically bound through covalent bonding with the MPDA and TMC.

This is explained in detail on pages 13 to 14, paragraph 3, lines 454 - 521, as follows:

“…In addition, the ZnO nanoparticle characteristic peaks were clearly visible on the XRD spectral analysis of the membranes modified with 1.50:1.00% ZnO/AA as shown in Figure 5 (a). This suggested higher concentration of ZnO nanoparticles available on the surfaces of this particular modified membrane (1.00% AA) than other modified membranes (1.50% AA) as shown in Figure 5 (a). The limited appearance of the ZnO nanoparticles characteristic peaks for the 1.5% AA modified membranes as shown in Figure 5 (b) was assigned to the higher concentration of AA (1.50%). The abundant presence of the AA could potentially shield the detection of ZnO nanoparticles when compared to the slightly lower 1.00% AA concentration. This could therefore suggest that the interaction between the 1.50% AA and ZnO nanoparticles inside the polyamide thin film matrices could potentially suppress the detection of the ZnO characteristic peaks, even at higher concentrations of the ZnO nanoparticles. Thus, the conclusion was that the ZnO nanoparticles were encapsulated into the AA matrices, making the nanoparticles more embedded into the polyamide thin film. This characteristic could further prevent leaching of the ZnO nanoparticles from the membranes.”

  1. Figure 2, scale bars and magnifications of SEM images should be inserted.

Answer: This has been corrected

  1. I suggest that the authors improve the quality of Figures 1, 4, 5, 6, 7, 8, 9 and 10, to clearly distinguish trends of each sample not only by different colours but also by different symbols and patterns.

Answer: This has been corrected and addressed as suggested.

  1. Page 10, lines 317-329, Figure 4 and 5 indicate results of XRD and TGA analysis for the samples. It does not represent hydrophilicity and permeability improvements by adding additives. Thus, these statements should be properly revised.

Answer: These statements have been revised as suggested. Here is the revised text, page 13, paragraph 2, lines 442 – 453:

“As shown in Figure 8 (a), the modified membranes (membranes modified with 1.00% AA and ZnO nanoparticles) were found to rather have higher permeability when compared to membranes modified with 1.5% AA and ZnO nanoparticles (Figure 8 (b)). However, according to Figure 5, the same membrane with higher water permeability, even though they are predominantly amorphous, the demonstrated increasing crystalline nature. As such, this contrasted the previous research that demonstrated that higher crystallinity translated into reduced water permeability [52–54]. This could be explained by considering that other membrane surface properties such as increasing roughness that could potentially promote the availability of water adsorption sites due to increased surface area. Furthermore, the modified membrane hydrophilicity contributed much more significantly into the water permeability performances than the increasing crystalline nature of the amorphous membranes.”

  1. Page 12, lines 393-395, again, Figure 4 does not represent permeability, so this should be revised.

Answer: This have been revised as suggested.

  1. Page 19, lines 616-619, there is a redundancy of ‘unmodified membranes’ should be revised.

Answer: This has been revised as suggested.

Reviewer 2 Report

This work investigates the effect of the incorporation of acrylic acid and ZnO nanoparticles in the polyamide layer of TFC membranes on the pH-responsive behaviour. The manuscript is well presented and could be published in MDPI membranes, after addressing the comments below.

1) The application of the responsive PA-TFC membranes reported in this work is not clear. Why is Pb(II) rejection tested? Is it a nanofiltration membrane? The difference of this work and the reported hydrogel-modified PA TFC membranes is not clear. Would be good to present a clear idea in the introduction to reflect novelty and practicability.

2) Fig 2: The scale bar is hard to read - please redraw the scale bar. It seems like the resolution is quite poor and the images are stretched horizontally. Please check and make sure the images are not distorted.

3) Fig 3: The image resolution needs to be improved.

4) Fig 4: The figures should be replotted- there should not be overlapping of the XRD spectrums. 

5) Fig 5, 6, 7, 8, 9 and 10: Poor resolution

6) The discussion of the rejection (from Line 645) is not clear. The pH value needs to be mentioned for obtaining 99.14% rejection. The discussion of the 'respective membranes' Line 646 is hard to follow.

7) The responsive behaviour of the membranes have not been reported. It would be good to show how the membranes respond to the changes to the pH in at least a few cycle in terms of flux and rejection. This will also indicate the stability of the membranes.

Author Response

Review 2

Comments and Suggestions for Authors

This work investigates the effect of the incorporation of acrylic acid and ZnO nanoparticles in the polyamide layer of TFC membranes on the pH-responsive behaviour. The manuscript is well presented and could be published in MDPI membranes, after addressing the comments below.

  • The application of the responsive PA-TFC membranes reported in this work is not clear. Why is Pb(II) rejection tested? Is it a nanofiltration membrane? The difference of this work and the reported hydrogel-modified PA TFC membranes is not clear. Would be good to present a clear idea in the introduction to reflect novelty and practicability.

Answer: in this research work, the Pb2+ pollutant was utilised as a model contaminant to test the strength of the pH responsive “gate effect” of the modified membranes. The aim was not only removing the Pb2+ through size exclusion, but clearly understand the adsorption-desorption mechanism as a result of the pH responsiveness of the modified membranes. Notably, most researchers graft the acrylic acid into commercial thin films. For example, pH-responsive poly(acrylic acid) (PAA) brushes were grafter on the surface of a commercial TFC-PA membrane using surface-initiated atom transfer radical polymerization (SI-ATRP). The NIPAmA copolymers and AA hydrogels were grafted on commercial PA-TFC membranes. However, the approach of this research work was to execute an in-situ incorporation of the AA during the interfacial polymerization process. Such an approach was envisaged to stabilize the AA and ZnO nanoparticles into the polyamide thin film matrix. Furthermore, the grafted PA-TFC membranes was often observed decreased water permeability because of the extra layer on the polyamide thin film. Therefore, in this research work, an in-situ interfacial polymerization was pursued.

  • Fig 2: The scale bar is hard to read - please redraw the scale bar. It seems like the resolution is quite poor and the images are stretched horizontally. Please check and make sure the images are not distorted.

Answer: This has been addressed

  • Fig 3: The image resolution needs to be improved.

Answer: This has been addressed

  • Fig 4: The figures should be replotted- there should not be overlapping of the XRD spectrums. 

Answer: This has been addressed

  • Fig 5, 6, 7, 8, 9 and 10: Poor resolution

Answer: These Figures have been addressed

6) The discussion of the rejection (from Line 645) is not clear. The pH value needs to be mentioned for obtaining 99.14% rejection. The discussion of the 'respective membranes' Line 646 is hard to follow.

This have been addressed seen line 705-713 page 22 and page 23 line 721. The highest rejection observed for the unmodified membrane was at pH 7. And the modified membranes showed high responsiveness of rejection at pH 11.

7) The responsive behaviour of the membranes has not been reported. It would be good to show how the membranes respond to the changes to the pH in at least a few cycles in terms of flux and rejection. This will also indicate the stability of the membranes

The responsiveness of the membranes has been well addressed on both permeability study (page 19, line 605-606) and rejection studies (see page 22 line 712-713). The change in the behaviour of the membrane in different pH levels proved the pH responsive effect of the membranes. However, the behaviour of the unmodified membrane to yield higher rejection at pH 7 cannot be well explained as it does not follow the “gate effect” as seen with the modified membranes.

Submission Date

29 September 2021

Date of this review

13 Oct 2021 04:47:00

Reviewer 3 Report

This manuscript reports polyamide-based membranes prepared with acrylic acids (AA) and ZnO particles. The effect of the loading of AA and ZnO on the physical properties and water permeability is investigated. The pH has a significant influence on water permeability and Pb rejection. The results are informative to the membrane and materials communities. More comments are shown below.

  1. What is the Pb rejection for typical RO or NF membranes?
  2. Equation 1 is missing
  3. Please explain the mechanism for acrylic acids (AA) to be incorporated into the PA structures. Does the AA react with MPDA? If so, how does it influence the reaction between MPDA and TMC?
  4. It is rather surprising to observe crystallinity for pure PA TFC and 1% AA samples (Figure 4a). This result is different from other results reported in the literature, to the best of my knowledge. Please provide a thorough comparison of this result with the literature.
  5. Please provide more details of the characterization of the selective layer, such as thickness, chemical compositions (the actual amount of AA and ZnO).
  6. What are the particle sizes of the ZnO?
  7. How does the pH value affect the film thickness?

Author Response

Review 3

Comments and Suggestions for Authors

This manuscript reports polyamide-based membranes prepared with acrylic acids (AA) and ZnO particles. The effect of the loading of AA and ZnO on the physical properties and water permeability is investigated. The pH has a significant influence on water permeability and Pb rejection. The results are informative to the membrane and materials communities. More comments are shown below.

  1. What is the Pb rejection for typical RO or NF membranes?

Answer: in this research work, the Pb2+ pollutant was utilised as a model contaminant to test the strength of the pH responsive “gate effect” of the modified membranes. The aim was not only removing the Pb2+ through size exclusion, but clearly understand the adsorption-desorption mechanism as a result of the pH responsiveness of the modified membranes. Notably, most researchers graft the acrylic acid into commercial thin films. For example, pH-responsive poly(acrylic acid) (PAA) brushes were grafter on the surface of a commercial TFC-PA membrane using surface-initiated atom transfer radical polymerization (SI-ATRP). The NIPAmA copolymers and AA hydrogels were grafted on commercial PA-TFC membranes. However, the approach of this research work was to execute an in-situ incorporation of the AA during the interfacial polymerization process. Such an approach was envisaged to stabilize the AA and ZnO nanoparticles into the polyamide thin film matrix. Furthermore, the grafted PA-TFC membranes was often observed decreased water permeability because of the extra layer on the polyamide thin film. Therefore, in this research work, an in-situ interfacial polymerization was pursued.

  1. Equation 1 is missing

Answer: This has been addressed

  1. Please explain the mechanism for acrylic acids (AA) to be incorporated into the PA structures. Does the AA react with MPDA? If so, how does it influence the reaction between MPDA and TMC?

Answer: The mechanism of the AA and MPDA was that they both react with TMC. As such, we gradually increased the AA concentration while at the same time decreasing the MPDA concentration to avoid competition towards the TMC, as shown in Table 1.

  1. It is rather surprising to observe crystallinity for pure PA TFC and 1% AA samples (Figure 4a). This result is different from other results reported in the literature, to the best of my knowledge. Please provide a thorough comparison of this result with the literature.

Answer: Notably, the modified and unmodified PA-TFC membranes were amorphous, which was consistent with literature reports. However, there was progressively increasing crystallinity, even though the membranes remained predominantly amorphous. The progressively increasing crystallinity of the amorphous membrane material from XRD spectra was estimated by using the intensity of the peaks and the shifting of the 2θ angles.

  1. Please provide more details of the characterization of the selective layer, such as thickness, chemical compositions (the actual amount of AA and ZnO).

Answer: The chemical compositions of the membranes constituted a polyamide thin film that was characteristic of the ZnO nanoparticles. This was demonstrated by the ATR-FTIR characterisation. However, the pH responsiveness of the membranes proved the AA presence in the polyamide thin layer. Again, from the, FTIR characterisation, the amide groups indicate the presence of polyamide layer onto the TFC membrane (see line 257, page 8). The actual amount of AA was (1 and 1.5 Wt%) and (0.5-1.5 Wt% of ZnO). The thickness of the membranes was not measured because it was irrelevant in this research work since the modified membranes changed thicknesses depending on the pH of the pollutant’s solution.

  1. What are the particle sizes of the ZnO?

Answer: the size of the ZnO was 40 nm

  1. How does the pH value affect the film thickness?

Answer: Swelling studies shows that membranes swell and shrink at the different pH levels. The swelling-shrinkage effect may be an indication that indeed the thickness of the membrane was affected at different pH levels. For example, at pH 11, the membrane’s thickness would increase because of swells due to the expansion of acrylic acid brushes. The expansion of the acrylic acid brushes led into an in increase in membrane thickness. While on pH 3, the AA brushes shrunk, resulting into low membrane thickness. (See page 20 line 626-628)

Submission Date

29 September 2021

Date of this review

10 Oct 2021 16:04:03

Round 2

Reviewer 3 Report

The authors have addressed my prior comments.